# DAG LEARNING ON THE PERMUTAHEDRON

**Valentina Zantedeschi**
Inria, Lille - Nord Europe research centre
The Inria London Programme, France and UK
University College London, Centre for AI
vzantedeschi@gmail.com

**Jean Kaddour, Luca Franceschi, Matt J. Kusner**
University College London, Centre for AI

**Vlad Niculae**
Informatics Institute, University of Amsterdam
v.niculae@uva.nl

## ABSTRACT

We introduce *Daguerro*, a strategy for learning directed acyclic graphs (DAGs). In contrast to previous methods, our problem formulation (i) guarantees to learn a DAG, (ii) does not propagate errors over multiple stages, and (iii) can be trained end-to-end without pre-processing steps. Our algorithm leverages advances in differentiable sparse structured inference for learning a total ordering of the variables in the simplex of permutation vectors (the permutahedron), allowing for a substantial reduction in memory and time complexities w.r.t. existing permutation-based continuous optimization methods.

## 1 INTRODUCTION

Learning a directed acyclic graph (DAG) from observational data is very challenging yet useful for many applications, e.g., in biology (Sachs et al., 2005), genetics (Zhang et al., 2013), or finance (Sanford & Moosa, 2012). The challenge comes from the combinatorial nature of the solution space, whose size grows super-exponentially with the number of variables $d$. While exact scored-based algorithms exist for small $d$ (Singh & Moore, 2005; Xiang & Kim, 2013; Cussens, 2011), approximate methods (Scanagatta et al., 2015; Aragam & Zhou, 2015; Ramsey et al., 2017) rely on global or local search heuristics in order to scale to problems with thousands of nodes. For instance, a large body of recent works (Zheng et al., 2018; Yu et al., 2019; Zheng et al., 2020; Ng et al., 2020; Brouillard et al., 2020; He et al., 2021) formulate DAG learning as a continuous optimization problem, where the acyclicity constraint is expressed as a smooth function and relaxed to allow efficient optimization with multi-purpose solvers. As such, the absence of cycles is no longer guaranteed and solutions often require post-processing.

Assuming identifiability of the underlying graph structure, another prominent line of works (Friedman & Koller, 2003; Gao et al., 2020; Reisach et al., 2021; Cundy et al., 2021; Charpentier et al., 2022) learns DAGs by (i) finding an ordering of the variables, and (ii) selecting the best scoring graph among (or marginalizing over) the structures that are consistent with the found ordering. Their benefit is to work on the space of orderings which is smaller and more regular than the space of DAGs (Friedman & Koller, 2003) while guaranteeing acyclicity. Existing works, however, are based either on two-steps procedures (Gao et al., 2020; Reisach et al., 2021) that do not guarantee optimality of the solution, or on end-to-end ones that involve higher computational costs (Friedman & Koller, 2003; Cundy et al., 2021; Charpentier et al., 2022). In particular, the recent work of Cundy et al. (2021) models a total ordering distribution on the Birkhoff polytope (the convex hull of permutation matrices) to obtain a differentiable operator, which has $O(d^3)$ time and $O(d^2)$ memory complexities. Charpentier et al. (2022) proposes using another operator with improved time complexity ($O(d^2)$) by constraining the permutation matrix to be row-stochastic (and not bistochastic).

In this work, we propose *Daguerro*, an end-to-end score-based strategy that belongs to the permutation-based family of works. Contrary to Cundy et al. (2021); Charpentier et al. (2022), we build our method on the SparseMAP operator (Niculae et al., 2018) for learning the total ordering of

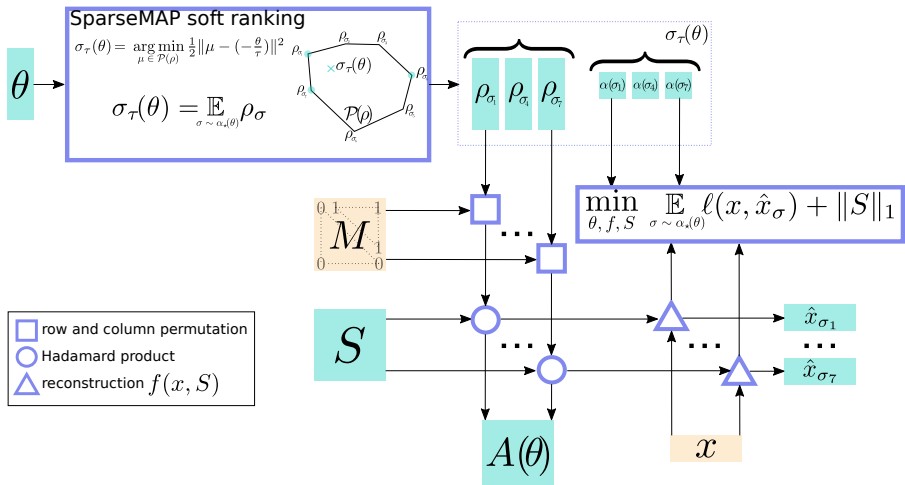

Figure 1: **Overview of *Daguerro*:** Our method learns the adjacency matrix $\mathbf{A}(\boldsymbol{\theta}) = \mathbf{S} \circ \mathbf{M}_{\sigma(\boldsymbol{\theta})}$, where we denote an observed data point by $x$, node utility scores by $\boldsymbol{\theta}$, a full and unconstrained weighted adjacency matrix by $\mathbf{S}$, parameterized permutations by $\sigma(\boldsymbol{\theta})$, and an upper strictly triangular masking matrix by $\mathbf{M}$.

the variables. This choice of differentiable operator allows us to work in the simplex of permutation vectors (a.k.a. Permutahedron) with significant gains in time ($O(d \log d)$) and space ($O(d)$) complexities (Blondel et al., 2020). Preliminary results on synthetic data show that *Daguerro* generally recovers the topological sort of the nodes but tends to return overly dense graphs.

## 2 METHOD

Let $\mathbf{X} \in \mathbb{R}^{n \times d}$ be a data matrix consisting of $n$ i.i.d. observations for $d$ nodes, where we denote the $j$-th column of $\mathbf{X}$ by $\boldsymbol{x}_j \in \mathbb{R}^n$, and $\mathcal{G} \in \mathbb{D}$ a DAG, where $\mathbb{D}$ denotes the discrete space of DAGs $\mathcal{G} = (\mathcal{V}, \mathcal{E})$ on $d$ nodes. We represent $\mathcal{G}$ by its adjacency matrix $\mathbf{A} \in \mathbb{R}^{d \times d}$, where an element $\mathbf{A}_{ij} \neq 0$ iff a directed edge exists from node $i$ to node $j$.

**Learning DAGs via ranking and masking** We formulate our score-based problem as follows

$$\min_{\boldsymbol{\theta}, \phi} \sum_{j=1}^{d} \ell \left( \boldsymbol{x}_j, f_j^{\phi} \left( \mathbf{X}, \mathbf{A}(\boldsymbol{\theta})_j \right) \right) \tag{1}$$
$$s.t.\ \mathbf{A}(\boldsymbol{\theta}) \in \mathbb{D},$$

where $\ell : \mathbb{R}^n \times \mathbb{R}^n \to \mathbb{R}$ is a point-wise loss function, $\{f^{\phi}\}_{j=1}^{d}$ is a set of functions parameterized by $\phi \in \mathbb{R}^{d_{\phi}}$ that computes $\boldsymbol{x}_j$ given its parents $pa(j) = \{i \in [d] \mid \mathbf{A}(\boldsymbol{\theta})_{ij} \neq 0\}$, and $\boldsymbol{\theta} \in \mathbb{R}^d$ is an utility vector whose role will be clarified later.

To enforce the constraint $\mathbf{A}(\boldsymbol{\theta}) \in \mathbb{D}$, we learn a total ordering of the variables and mask out all edges that are not consistent with it: if $\boldsymbol{x}_j \prec \boldsymbol{x}_i$ no edge can be drawn from $i$ to $j$. As this procedure returns only transitive closures, we further encourage the removal of unnecessary edges via $l_1$ regularization (as later defined in eq. (7)).

We decompose $\mathbf{A}(\boldsymbol{\theta})$ as follows:

$$\mathbf{A}(\boldsymbol{\theta}) = \mathbf{S} \circ \mathbf{M}_{\sigma(\boldsymbol{\theta})}, \tag{2}$$

where $\mathbf{S} \in \mathbb{R}^{d \times d}$ is a full and unconstrained weighted adjacency matrix, $\sigma(\boldsymbol{\theta}) \in \Sigma_d$ is a parametrized permutation with $\Sigma_d$ being the set of all $d$-permutations, and $\mathbf{M} \in \{0, 1\}^{d \times d}$ is an upper strictly triangular masking matrix such that $\mathbf{M}_{ij} = 1$ if $i < j$ and $0$ otherwise. Further, $\circ$ denotes the Hadamard product, and $\mathbf{M}_{\sigma(\boldsymbol{\theta})}$ indicates that the rows and columns of $\mathbf{M}$ have been permuted according to $\sigma(\boldsymbol{\theta})$.

**Soft ranking on the permutahedron.** In contrast to Cundy et al. (2021) that operates in the Birkhoff polytope, to learn the optimal permutation we turn to an efficient formulation of ranking as constrained optimization on the permutahedron, following Blondel et al. (2020). The ranking of a vector $\boldsymbol{\theta} \in \mathbb{R}^d$, i.e., the permutation $\sigma \in \Sigma_d$ that arranges the elements of $\boldsymbol{\theta}$ in decreasing order, can be written as (Blondel et al., 2020, Lemma 1)

$$\sigma(\boldsymbol{\theta}) = \arg\max_{\sigma \in \Sigma_d} \langle \boldsymbol{\theta}, \boldsymbol{\rho}_\sigma \rangle, \quad \text{where } \boldsymbol{\rho} = (d, d-1, \dots, 1). \tag{3}$$

The permutahedron of a vector $\boldsymbol{w} \in \mathbb{R}^d$ is the convex hull of all possible permutations of $\boldsymbol{w}$:

$$\mathcal{P}(\boldsymbol{w}) = \text{conv}\{\boldsymbol{w}_\sigma : \sigma \in \Sigma_d\}. \tag{4}$$

Blondel et al. (2020) show that projection onto $\mathcal{P}(\boldsymbol{\rho})$ yields a soft ranking operator:

$$\sigma_\tau(\boldsymbol{\theta}) := \arg\min_{\boldsymbol{\mu} \in \mathcal{P}(\boldsymbol{\rho})} \frac{\tau}{2} \|\boldsymbol{\mu} - (-\boldsymbol{\theta})\|^2 = \arg\min_{\boldsymbol{\mu} \in \mathcal{P}(\boldsymbol{\rho})} \frac{1}{2} \|\boldsymbol{\mu} - (-\boldsymbol{\theta}/\tau)\|^2. \tag{5}$$

This mapping is differentiable w.r.t. $\boldsymbol{\theta}$ and, in the absence of ties, as the regularization parameter $\tau$ goes to zero, $\sigma_\tau(\boldsymbol{\theta}) \to \sigma(\boldsymbol{\theta})$ (Blondel et al., 2020, Proposition 2). An algorithm based on isotonic regression permits $O(d \log d)$ computation of $\sigma_\tau(\boldsymbol{\theta})$. However, soft rankings are not suitable for our goal, since we cannot replace $\mathbf{M}_{\sigma(\boldsymbol{\theta})}$ by $\mathbf{M}_{\sigma_\tau(\boldsymbol{\theta})}$. In fact, the latter is only defined for the vertices of the permutahedron, i.e., discrete permutations $\rho \in \Sigma_d$.

The same problem arises when working with soft permutation matrices, as done in Cundy et al. (2021) and Charpentier et al. (2022). When leveraging Mena et al. (2018)'s work, the authors resort to the straight-through gradient estimator (Bengio et al., 2013), which boils down to (i) finding a hard permutation matrix in the forward pass by solving an optimal matching program via the Hungarian algorithm (Kuhn, 2010), and (ii) computing the gradients of its continuous approximation obtained through the Sinkhorn operator (Sinkhorn, 1964) for the backward pass. The straight-through gradient estimator is also deployed when leveraging the SoftSort operator (Prillo & Eisenschlos, 2020), as done in (Charpentier et al., 2022) where the one-hot encoded argsort is used for the forward pass. This strategy however induces additional bias in the training procedure.

**SparseMAP distributions over permutations.** To handle the issue that our loss can only be evaluated meaningfully at discrete permutations, we note that any $\boldsymbol{\mu} \in \mathcal{P}(\boldsymbol{\rho})$ can be written as a convex combination of vertices,

$$\boldsymbol{\mu} = \sum_{\sigma \in \Sigma_d} \alpha(\sigma) \boldsymbol{\rho}_\sigma = \mathbb{E}_{\sigma \sim \alpha}[\boldsymbol{\rho}_\sigma], \quad \text{where } \alpha(\cdot) \geq 0, \text{ and } \sum_{\sigma \in \Sigma_d} \alpha(\sigma) = 1. \tag{6}$$

SparseMAP (Niculae et al., 2018) is a strategy for differentiable structured inference that solves problems of the form of eq. (5), yielding not only the optimal solution $\sigma_\tau(\boldsymbol{\theta}) = \boldsymbol{\mu}_*$, but also a sparse decomposition $\alpha_*(\boldsymbol{\theta})$, such that $\boldsymbol{\mu}_* = \mathbb{E}_{\sigma \sim \alpha_*}[\boldsymbol{\rho}_\sigma]$. This is achieved via an active set algorithm, which maintains a sparse decomposition, and iteratively adds or removes permutations based on linear oracles of the form (3), implemented by a sorting algorithm. This iterative algorithm is more costly than the direct isotonic regression one, but provides the decomposition $\alpha_*$, which allows us to compute exact marginalizations of the form $\mathbb{E}_{\sigma \sim \alpha_*(\boldsymbol{\theta})}[f(\sigma)]$ by explicitly summing over the sparse set of permutations found by SparseMAP, as proposed for structured latent variable learning by Correia et al. (2020).

**DAG learning on the permutahedron.** Our final bi-level optimization problem is:

$$\min_{\boldsymbol{\theta}, \mathbf{S}, \phi} \mathbb{E}_{\sigma \sim \alpha_*(\boldsymbol{\theta})} \left[ \sum_{j=1}^d \ell\left(\boldsymbol{x}_j, f_j^\phi(\mathbf{X}, \mathbf{S} \circ \mathbf{M}_\sigma)\right) \right] + \lambda \|\mathbf{S}\|_1 \tag{7}$$

$$s.t. \; \mathbb{E}_{\sigma \sim \alpha_*(\boldsymbol{\theta})}[\boldsymbol{\rho}_\sigma] = \arg\min_{\boldsymbol{\mu} \in \mathcal{P}(\boldsymbol{\rho})} \frac{1}{2} \|\boldsymbol{\mu} - (-\boldsymbol{\theta}/\tau)\|^2 \tag{8}$$

The inner problem eq. (8) is optimized via SparseMAP, which returns a sparse categorical distribution $\alpha_*$ and its gradients w.r.t. $\boldsymbol{\theta}$. We can solve the outer optimization via proximal adaptive gradient methods (Parikh & Boyd, 2014). A schematic of the overall approach is presented in Figure 1.

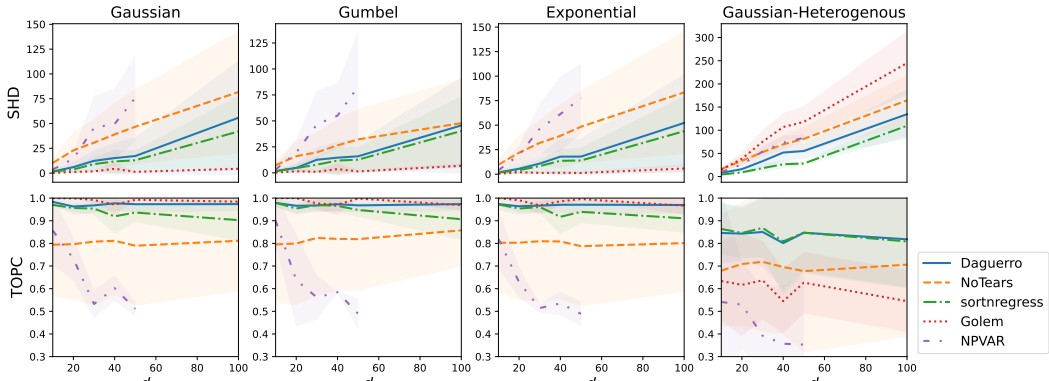

Figure 2: Structural Hamming Distance (SHD, the lower the better) and Topological Ordering Pearson Correlation (TOPC, the higher the better) as a function of the number of nodes $d$ and for three different noise models (in column) and $n = 1000$ training points (which are not enough to run *NPVAR* with $d = 100$). We average results over the three graph models, where for each setting we ran the methods on 5 different datasets. Detailed results in Appendix A.

**Computational details.** Each SparseMAP iteration involves a length-$d$ argsort and a Cholesky update of a $s$-by-$s$ matrix, where $s$ is the size of the active set, bounded by the iteration number. With a constant number of iterations (as in our implementation), this leads to an overall time complexity of $O(sd\log d + s^2)$ and space complexity $O(sd)$. If run until exact convergence, Carathéodory's convex hull theorem guarantees $s \leq d + 1$ leading to cubic worst-case complexity; however, in practice, we almost always reach convergence within 100 iterations. In addition, we warm-start the sorting algorithm with the last selected permutation. This is better both in theory and in practice than the $O(d^3)$ complexity of maximization over the Birkhoff polytope. A tighter analysis or specialized algorithm based on isotonic regression is a promising direction for future work.

## 3 EXPERIMENTS

We report a set of preliminary experiments to validate the proposed method for learning DAGs from observational data. We consider synthetic settings with simulated linear additive noise models. Following common practices from the literature (Zheng et al., 2018; Gao et al., 2020), we generate a random graph with $2*d$ expected edges according to the Erdös-Rényi, Scale-Free or BiPartite models and assign to each edge a weight uniformly drawn in $[-2, -0.5] \cup [0.5, 2]$. We then sample data according to the linear model $XA + \varepsilon$, where $\varepsilon \in \mathbb{R}^d$ is an exogenous random variable whose elements are independently distributed as Normal$(0, 0.5)$ (**Gaussian**), Normal$(0, \sigma_j)$ with $\sigma_j \in [0, 0.5]$ (**Gaussian-Heterogeneous**), Exponential$(0.5)$ (**Exponential**) or Gumbel$(0, 0.5)$ (**Gumbel**).

As baselines, we consider state-of-the-art approximate score-based methods: *NoTears* (Zheng et al., 2018), the first continuous optimization method, which optimizes the Frobenius reconstruction loss and where the DAG constrain is enforced via the Augmented Lagrangian approach; *Golem* (Ng et al., 2020), another continuous optimization method which optimizes the data likelihood (under Gaussian equal variance error assumptions) regularized by *NoTears*'s DAG penalty; *NPVAR* (Gao et al., 2020), an iterative algorithm that learns topological layers and then prunes edges based on node residual variance (with Generalized Additive Models regressor backend to estimate conditional variance); *sortnregress* (Reisach et al., 2021), another two-steps strategy that orders nodes by increasing variance and selects the parents of a node among all its predecessors using the Least Angle Regressor (Efron et al., 2004). Before evaluation, we post-process the graphs found by *NoTears* and *Golem* by iteratively removing edges ordered by increasing weight until obtaining a DAG, as they often contain cycles. For our method, we instantiate $f_j^\phi(\mathbf{X}, \mathbf{A}(\boldsymbol{\theta})_j) = \mathbf{X}\mathbf{A}(\boldsymbol{\theta})_j$ and optimize the data likelihood (under Gaussian equal variance error assumptions).

We set the hyper-parameters of all methods to their default values, apart from *sortnregress* that uses the Bayesian Information Criterion for model selection. In particular, for *Daguerro* we set $\lambda = 0.1$,

$\tau = 1$ and remove all edges with weights $\leq 0.3$ as commonly done in the literature. We additionally apply a $l_2$ regularization to all learned parameters (and set its hyper-parameter to $0.01$) as we find it stabilizes training.

In Figure 2, we compare the methods by two metrics: the Structural Hamming Distance (*SHD*) between true and estimated graphs, as standard in the literature; and the Topological Ordering Pearson Correlation (*TOPC*) that we introduce to measure how well a method is able to estimate the true topological layers of the nodes. We believe it is essential to study both metrics because the *SHD* alone does not distinguish between wrong edges that are consistent with the variable ordering from wrong edges that are not. For instance, the transitive closure of a sparse graph has high *SHD* even though it preserves the true ordering of the nodes.

*TOPC* computes the Pearson's correlation coefficient between the true topological layers and the learned ones. Formally, given a DAG $\mathcal{G} = (\mathcal{V}, \mathcal{E})$ and its adjacency matrix $\mathbf{A}$ we define the topological layers ( or layer decomposition) $L(\mathcal{G}) = (L_1, \ldots, L_R)$ following Gao et al. (2020), where $L_{\mathbf{A}} = \{\boldsymbol{x}_j \mid pa(j) \in (\cup_{r'=1}^{r-1} L_{r'}) \wedge \boldsymbol{x}_j \in (\mathcal{V} \setminus \cup_{r'=1}^{r-1} L_{r'})\}$ is the set of nodes that are sources of the DAG $\mathcal{G}[\mathcal{V} \setminus \cup_{r'=1}^{r} L_{r'}]$ and $1 \leq R \leq d$ is the depth of the graph. Note that this layer decomposition is unique, as any node $\boldsymbol{x}_j$ belongs to a single layer. We can then assign to each node $\boldsymbol{x}_j$ the index of the layer it belongs to, obtaining the assignment vectors $\boldsymbol{r}_{\mathbf{A}}$. *TOPC* between estimated $\hat{\mathbf{A}}$ and true $\mathbf{A}$ adjacency matrices is computed as

$$TOPC(\hat{\mathbf{A}}, \mathbf{A}) = \frac{\text{Cov}(\boldsymbol{r}_{\hat{\mathbf{A}}}, \boldsymbol{r}_{\mathbf{A}})}{\sqrt{\text{Var}(\boldsymbol{r}_{\hat{\mathbf{A}}})}\sqrt{\text{Var}(\boldsymbol{r}_{\mathbf{A}})}}$$

with $\text{Cov}(\cdot)$ denoting the covariance and $\text{Var}(\cdot)$ denoting the variance.

In terms of *TOPC*, we observe that *Daguerro* provides better orderings than those of *NoTears* and *NPVAR*; it sightly improves over *sortnregress* and it is on-par with *Golem* on the equal-variance noise models, while clearly improving on *Golem* in the **Gaussian-Heterogeneous** setting. In terms of *SHD*, we find that *Daguerro* never provides the best performance, although it still improves over *NoTears* and *NPVAR*. Given these results, we believe that a better strategy for pruning edges is key to improve our method. Additional improvement may come from tuning our hyper-parameters.

## 4 CONCLUSION AND FUTURE WORK

In this work, we presented *Daguerro*, an efficient score-based method for end-to-end learning of directed acyclic graphs and structural equation models. *Daguerro* reliably finds correct total ordering, yet tends to yield too dense graphs. We anticipate that incorporating a $l_0$ regularization term (Louizos et al., 2018) could help mitigate this latter issue. Future directions include adapting our strategy to non-linear SEMs (less amenable to continuous optimization approaches) and experimenting with interventional data. While our current implementation uses SparseMAP, other recent strategies for structured discrete distributions such as SST (Paulus et al., 2020) or I-MLE (Niepert et al., 2021) could offer compelling alternatives to optimize the structure parameters.

### ACKNOWLEDGEMENTS

We thank the anonymous reviewers for their constructive feedback. VZ is supported by the French Project APRIORI ANR-18-CE23-0015. JK acknowledges support by the Engineering and Physical Sciences Research Council with grant number EP/S021566/1. VN is partially supported by the Hybrid Intelligence Centre, a 10-year programme funded by the Dutch Ministry of Education, Culture and Science through the Netherlands Organisation for Scientific Research (https://hybrid-intelligence-centre.nl).

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

## A  DETAILED RESULTS

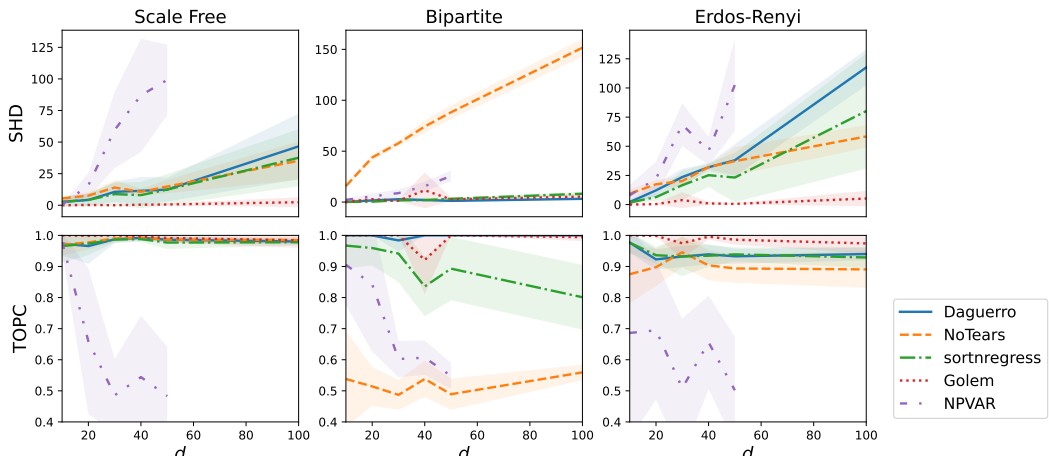

Figure 3: Detailed results with *Gaussian* noise model.

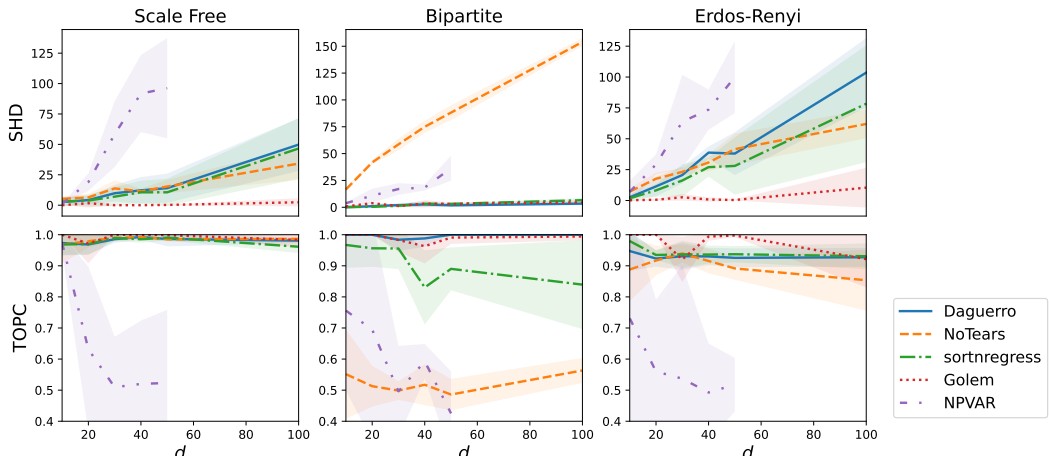

Figure 4: Detailed results with *Exponential* noise model.

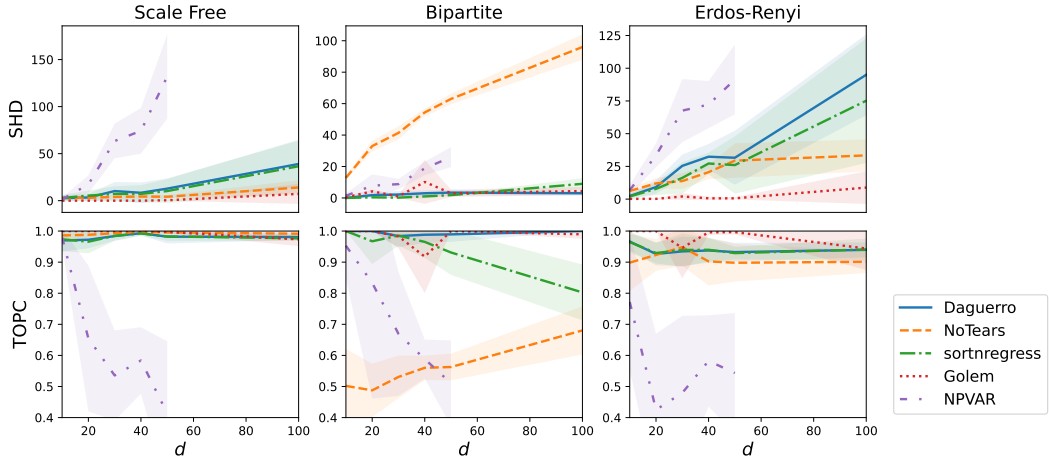

Figure 5: Detailed results with *Gumbel* noise model.

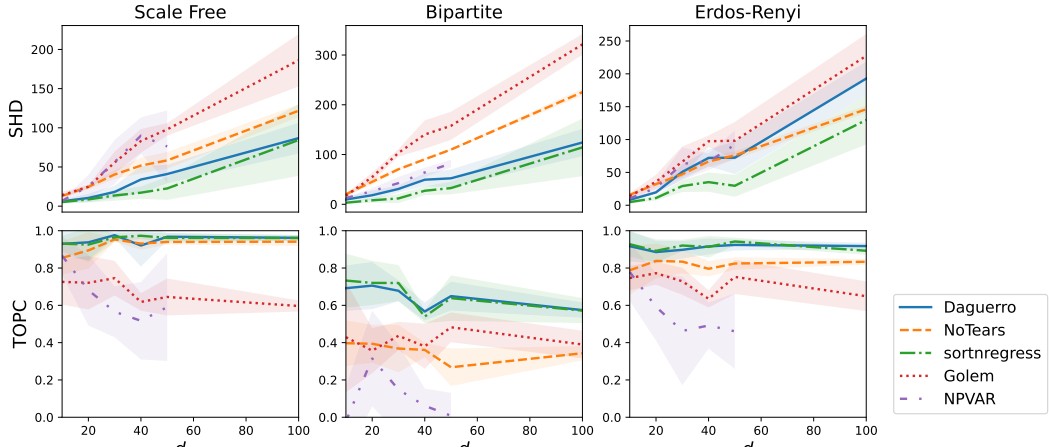

Figure 6: Detailed results with *Gaussian-Heterogeneous* noise model.

