# OpenReview forum: "DAG Learning on the Permutahedron"
_ICLR.cc/2022/Workshop/OSC — ICLR2022 OSC  Poster_

### Official Review · Reviewer_WGNA · 2022-03-02
**Review. Briefly: A Positive Contribution to the DAG Learning Literature**

**Rating:** 3
**Confidence:** 3

**Review:**

**1. Summary and Contributions**

The paper considers considers causal discovery (or induction), that is, learning the underlying causal relations of our data's variables using said data. Specifically, it builds upon several previous works including continuous optimization and soft ranking for DAG learning, revealing a new method that leverages the computational advantages of taking a permutahedron perspective. The paper provides an empirical investigation justifying the newly proposed bi-level optimization formalism and corroborating on the proposed time and memory complexities.

**2. Strengths**

The paper has several strengths, considered one-by-one in the following list (the list is ordered in correspondence to the paper presentation):

* The grand problem of causal discovery is essential to human cognition and thus to science and engineering, and all its instances, and tackling specific sub-problems - as done in this work - is determining for developing next generation learning systems.
* The guarantee of a directed, acyclic graph (DAG) by construction opposed to pragmatically obtaining it through heuristic procedures.
* A comprehensive and extensive landscape of many relevant works for DAG learning is being presented.
* They propose quasi-linear and linear time and memory complexities respectively (at least practically for the investigated settings).
* The notation is being compactly presented a-priori, in support of the overall clear manner of presentation.

**3. Weaknesses**

The paper suffers from several disadvantages (ranging in importance from minor to more fundamental) that however IMHO can be improved upon mostly quickly. Thereby, the following list - again one-by-one - aims to provide specific pointers with improvement suggestions if applicable (please note, the list is unordered):

* To avoid clutter in the presentation for instance when re-citing certain works several times within the introduction on p.1, the authors might want to consider directly referencing key aspects of a given reference.
* The presentation of the proposed formalism (Eqs.7,8) might significantly benefit from for instance color highlighting alongside a corresponding legend when explaining components. It might also significantly benefit from an improvement of the key Figure (Fig.1) using for instance a comprehensive caption or by extending it with an illustrative/motivating example (since as of now, the figure only provides a notion of dependency/order within the components of the approach). Furthermore, the authors might want to consider referencing the key Figure earlier than p.3 (since Fig.1 appears on p.2).
* Reisach et al.'s main claim is that using simulated DAGs with score-based methods like NOTEARS or Golem is misleading since standardizing the data leads to degenerated performances at the same level of random guessing (suggesting that said methods exploit the sortability of the variance of the data). Since this is an arguably important aspect to be discussed within the community (since it has major consequences for the future of this research direction), the authors should consider elaborating whether their proposed approach can counteract difficulties that might be in common with the formerly mentioned methods.
* Consider writing $\mathbb{R}^n$ as the domain for $l$ in Eq.1
* Consider removing the $d$ in $d < i < j$ for Eq.2
* Since time complexity is a severely limiting factor in methods (like NOTEARS) that make use of a continuous acyclicity constraint as they scale (e.g. cubically) in the number of nodes (where we might observe in relevant, real-world settings like social networks large numbers of nodes), the authors should consider providing a more detailed, rigorous analysis of asymptotic time complexity.

**3. Correctness, Clarity, and Literature**

No contradictions or any sort of relevant mistake have been detected in the paper. The clarity of the paper is an advantage. Existing bodies of work are being referenced accordingly at the end of the paper.

**4. Reproducibility, Code Release, and Assumptions**

Sufficient details for reproduction are being provided. Unfortunately, without actual code. All key assumptions for the method are being pointed out explicitly.

---

### Official Review · Reviewer_8meV · 2022-03-16
**Review for DAG learning of the permutahedron**

**Rating:** 2
**Confidence:** 3

**Review:**

This paper builds on the work of "Fast Differentiable Sorting and Ranking" for DAG learning.

Here are my comments for the paper.

1. The idea is interesting, however, the writing of the paper could be improved.
2. There is a very related paper "Differential DAG sampling", which also learns a DAG by first learning the ordering of the nodes (permutation) and then learning an upper triangular matrix. It would be nice if the authors can discuss and compare to this paper.
3. The procedure used by the algorithm seems to reject/ mask out edges that are not consistent with the permutation (ordering) of the nodes. I am curious to see what is the percentage of the edges that are rejected or masked out.
4. The model is trained on observational data, hence, the underlying model can only discover up to a Markov equivalence class of graphs. I am curious to see if the model can indeed learn different modes, aka, can it sample different node pertmuations that are consistent with graphs within the same markov equivalence class.
5. The authors also mentioned that they introduced the  Topological Ordering Pearson Correlation evaluation metric to evaluate the permutation (node ordering) the model has learned, but I did not see a description of this metric in the paper. Could the authors elaborate on that?

Overall, I think this is an interesting idea, but the writing of the paper can benefit from some more work. Also, it would be interesting to see the ablations studies and also comparisons to related methods that was discussed earlier.

---

### Decision · Program_Chairs · 2022-03-23

Accept (Poster)